# Fracture Analysis of Particulate Metal Matrix Composite Using X-ray Tomography and Extended Finite Element Method (XFEM)

**Rui Yuan** [1], **Sudhanshu S. Singh** [1,2], **Xiao Liao** [1], **Jay Oswald** [1,*] and **Nikhilesh Chawla** [1,*]

[1]  School for the Engineering of Matter, Transport and Energy, Arizona State University, Tempe, AZ 85287-6106, USA; yuanrui.pc@hotmail.com (R.Y.); sudhanss@iitk.ac.in (S.S.S.); jimliao4@gmail.com (X.L.)

[2]  Department of Materials Science and Engineering, Indian Institute of Technology Kanpur, Kanpur UP-208016, India

\*  Correspondence: joswald1@asu.edu (J.O.); nchawla@asu.edu (N.C.); Tel.: +480-965-4317 (J.O.); +480-965-2402 (N.C.)

**Abstract:** Particle reinforced metal matrix composites (MMCs) offer high strength, low density, and high stiffness, while maintaining reasonable cost. The damage process in these MMCs starts with either the fracture of particles or by the de-cohesion of the particle-matrix interfaces. In this study, the extended finite elements method (XFEM) has been used in conjunction with X-ray synchrotron tomography to study fracture mechanisms in these materials under tensile loading. The initial 3D reconstructed microstructure from X-ray tomography has been used as a basis for the XFEM to simulate the damage in the 20 vol.% SiC particle reinforced 2080 aluminum alloy composite when tensile loading is applied. The effect of mesh sensitivity on the Weibull probability has been studied based on a single sphere and several particles with realistic geometries. Additionally, the effect of shape and volume of particles on the Weibull fracture probability was studied. The evolution of damage with the applied traction has been evaluated using simulation and compared with the experimental results obtained from *in situ* tensile testing.

**Keywords:** metal matrix composite; extended finite element method; X-ray tomography; Weibull fracture probability

## 1. Introduction

Metal matrix composites (MMCs) are attractive for many applications due to their excellent properties, including high strength and low density [1]. Advances in material processing make it possible to fabricate advanced composites with customized geometries and multifunctional properties; however, for material design and optimization, a more thorough understanding of the relationship between microstructure and properties is needed. Computational simulations can help in revealing and quantifying the influence of microstructure on properties, such as strength and stiffness. Traditional finite element and computed aided design software packages, using parametric geometrical descriptions, are unwieldy and possess great difficulty in handling complex geometries. As a result, the analysis of particle reinforced MMCs has been largely limited to either two-dimensional models, and/or with simplified shapes of reinforcing particles, e.g., spheres or ellipsoids [2–5]. Furthermore, while such idealized microstructural models can be used to predict composite properties that reflect the average microscopic material response, they are not sufficient to describe phenomena governed by extreme values. For instance, microscopic stress concentrations that lead to the nucleation of cracks are strongly influenced by the size, shape, and distribution of reinforcing particles, and, thus, will not be accurately represented in models with simplified geometry.

Three-dimensional (3D) characterization methods, such as X-ray synchrotron tomography, can provide a wealth of data characterizing microstructural features in statistically significant volumes [6]. X-ray tomography has been successfully applied to characterize the microstructures in 3D for materials, such as metal matrix composites [7–9], Al alloys [10,11], Sn-rich alloys [12] and Magnesium alloys [13]. Due to its non-destructive nature, in situ X-ray synchrotron experiments have also been conducted to understand the deformation behavior in real-time (4D), such as fatigue [14–16] and stress corrosion cracking (SCC) [14,17]. Furthermore, these tomography experiments can be paired with the simulations of the same microstructure in order to calibrate and/or validate models [18,19]. Waton et al. [20], proposed a finite element method for the simulation of mechanical properties of two phase systems. However, the complexity of most real material geometries compounded by the immense size of microstructural data sets pose a significant modeling challenge for the traditional finite element codes, namely that it becomes increasing difficult to generate high quality meshes where element faces conform to the material interfaces.

To overcome these challenges, extended finite element method (XFEM) was developed by Belytschko et al. [21,22] to model crack propagation without the need to remesh as the crack propagates. The central idea of the XFEM is to enrich the finite element basis with functions that locally represent the solution, e.g., discontinuous functions are injected to the approximation space along crack surfaces. The enrichments are introduced by a partition of unity [23], typically using the regular finite element shape functions to blend the enrichment functions into the approximation of the solution. Huynh and Belytschko [24] presented further developments in XFEM for fracture problems in composite materials, where fracture along material interfaces was modeled by combining both strong and weak discontinuities, i.e., discontinuities in strain and displacement, respectively. Ye et al. [25] implemented the XFEM within an ABAQUS subroutine to study the influence of reinforcing particles on the crack propagation behavior in a MMC. Wang et al. [26] investigated the interaction between a propagating crack and single or multiple particles in a brittle matrix.

Failure strengths of brittle materials vary unpredictably over a wide range from specimen to specimen even though they are manufactured in the same way and tested under the same condition [27,28]. Therefore, fracture statistics have to be applied to understand the failure strengths of brittle reinforced particles in MMCs. The Weibull distribution [29] can be used to evaluate the failure probability of the brittle reinforced particles. The Weibull distribution has been applied to many problems, including the study of yield strength of pentagonal silver nanowires [30], the modeling of thermal inactivation of microbial vegetative cells [31], the fatigue life prediction based on crack growth data [32], etc. Further, the Weibull distribution has been proven to be a suitable empirical statistical distribution for cleavage fracture in brittle materials [33–36]. Eckschlager et al. [37] proposed a finite element based approach for modeling brittle cleavage of the spherical particles on the basis of Weibull fracture probabilities. Doremus [27] compared normal, Weibull and Type I extreme value distributions for failure strengths of glass. Lu et al. [28] fitted fracture strength data to Weibull and normal distributions for three types of brittle materials and showed that the difference between the two distributions was very small to be clearly distinguished in the case of SiC. Using the fracture toughness data under both low and high constraint conditions at the crack front, Gao et al. [38] proposed a new way to calibrate Weibull stress parameters analytically. In another paper by Gao et al. [39], a new strategy was used to calibrate the Weibull stress model to predict the cleavage fracture in plates containing surface cracks.

It is clear that adequate visualization and fracture quantification are critical to the understanding of damage in MMCs. Therefore, in this work, XFEM has been used in combination with in-situ X-ray synchrotron tomography to understand the fracture of brittle particles in particulate reinforced MMCs using a Weibull distribution model. A systematic and microstructure-based understanding of damage and fracture in these materials was obtained and is discussed.

## 2. Methodology

### 2.1. Implicit Geometry Representation

In a heterogeneous material, each three-dimensional phase can be approximated as a set of voxels contained within the phase as:

$$v_{ijk}^{\alpha} = \begin{cases} 1 & \text{if } \ x_{ijk} \ \epsilon \ \Omega_{\alpha} \\ 0 & \text{else} \end{cases} \tag{1}$$

where, $\Omega_{\alpha}$ denotes the domain for the $\alpha$ phase. The location of each voxel is described by a 3-tuple of integers such that:

$$x_{ijk} = x_0 + i \, \Delta_x + j \, \Delta_y + k \, \Delta_z \tag{2}$$

where, $x_0$ is the origin of the voxel data set. The voxel cell length in three directions is determined by the lengths of the voxel cell vectors, i.e., $\|\Delta_x\|$, $\|\Delta_y\|$, and $\|\Delta_z\|$.

In our previous work [40], we have developed a geometry segmentation algorithm to accurately identify and separate discrete geometric features. In this algorithm, the betweenness centrality, which is a measure of the importance of a node with respect to the connectivity of a network, is used to identify voxels that create spurious bridges. To facilitate the automation of the new algorithm, we developed a non-dimensional relative centrality metric to allow for the selection of a threshold criteria that was independent of inclusion shape or volume.

In this work, level sets are employed to represent microstructures implicitly. Level set methods provided a concise way to describe complex microstructures with consistent mesh quality and level set fields could be conveniently applied in enrichments. In the earlier studies, level sets have already been introduced for material interface modeling [41,42] and crack modeling [43,44]. The phase interfaces are represented by the zero-level set of a continuous level set function as:

$$f_{\alpha}^{int}(x) = 0, \ \alpha = 1 \dots n^{int} \tag{3}$$

where, $n^{int}$ is the number of reinforcing phases. For each phase, one level set can represent all interfaces of that phase. If point x is inside the $\alpha$ phase, $f_{\alpha}^{int}(x)$ will be negative, otherwise, it will be positive. Recently, we have presented two methods for level set initialization of complex material interfaces [45]. In the first method, a level set evolution equation was formulated and solved by the Galerkin method. In the second approach, the distance field was initialized by the fast-marching method [46,47] on a uniform grid, and then the solution was projected onto the finite element mesh by least squares. The second approach was found to be superior in speed and accuracy, which is also applied to initialize the matrix/reinforcement interfaces in the current work.

To represent the crack surface on a fractured particle, a similar level set function is introduced for each particle as:

$$f_{\beta}^{cr}(x) = 0, \quad \beta = 1 \dots \dots n^{par} \tag{4}$$

where, $\beta$ is a particle id and $n^{par}$ the total number of particles, and the zero level set gives the crack surface of the particle. Note that the above level set field defines a surface that spans the entire simulation domain. In order to restrict the crack surfaces to within the particle, one more level set function is used to define the location of the crack tip as:

$$g_{\beta}^{cr}(x) < 0, \ \text{if x is inside the particle } \beta \tag{5}$$

$$g_{\beta}^{cr}(x) > 0, \text{if x is outside the particle } \beta \tag{6}$$

$$g_{\beta}^{cr}(x) = 0, \ \text{if x is at the interface of the particle } \beta \tag{7}$$

Therefore, the crack surface on the particle β can be represented by the combination of two level set fields $f_{\beta}^{cr}(x) = 0$ and $g_{\beta}^{cr}(x) \leq 0$.

### 2.2. Formulation of XFEM for Discontinuities

In traditional finite element models, discontinuities must reside along element faces such that the discontinuities can be represented explicitly in finite element models. However, in XFEM, the interior discontinuities are represented implicitly by the level set fields and the displacement field approximates by a discontinuous displacement enrichment [21] based on a local partition of unity [23]. Given a finite element model $\Omega \in \mathbb{R}^3$, partitioned into finite elements, let $S$ be the set of all finite element nodes, $S^{cr}$ be the set of nodes of elements whose edges are intersected by a crack surface $\Gamma^{cr}$, and $S^{int}$ be the set of nodes of elements intersected by the material interface $\Gamma^{int}$. The XFEM displacement field can be expressed by:

$$\mathbf{u}^h(\mathbf{x},t) = \sum_{I \in S} N_I(\mathbf{x})\mathbf{u}_I + \sum_{J \in S^{cr}} \phi_J(\mathbf{x})\mathbf{b}_J + \sum_{K \in S^{int}} N_K(\mathbf{x})\psi(\mathbf{x})\mathbf{q}_K \tag{8}$$

where, $\mathbf{u}_I$ and $N_I(\mathbf{x})$ are the nodal displacements and finite element shape functions, respectively. Additional enriched degrees of freedom $\mathbf{b}_J$ and $\mathbf{q}_K$ are for crack and material interfaces, respectively. The function $\phi(\mathbf{x})$ represents a jump enrichment which introduces a discontinuity in the displacement at the crack surface. The function $\psi(\mathbf{x})$ represents a kink enrichment, which introduces a discontinuity in the gradient of the displacement at a material interface. For a strongly-bonded material interface, the displacement field remains continuous; however, strain can be discontinuous on the interface. The jump enrichment function for a crack is given by [48]:

$$\phi_J(\mathbf{x}) = N_J(\mathbf{x})\left[H(f^{cr}(\mathbf{x})) - H\left(f^{cr}(\mathbf{x}_J)\right)\right]H\left(-g_\beta^{cr}(\mathbf{x})\right) \tag{9}$$

where, $H(\cdot)$ is the Heaviside step function given by:

$$H(\mathbf{x}) = \begin{cases} 1 & if \quad \mathbf{x} > 0 \\ 0 & \qquad else \end{cases} \tag{10}$$

The simplest kink enrichment function is an absolute value function [41,42,49]. Modeling the interfaces with the absolute value function is troublesome since it does not vanish at the edges of the elements intersected by the interfaces. Moës et al. [50] proposed a modified kink enrichment function to preserve the ridge at the interfaces, but also vanishes at the edges of enriched elements. This enrichment function is given as:

$$\psi(\mathbf{x}) = \sum_I \left|f^{int}(\mathbf{x})\right|N_I(\mathbf{x}) - \left|\sum_I f^{int}(\mathbf{x})N_I(\mathbf{x})\right| \tag{11}$$

This enrichment function eliminates the need of blending elements such that only elements intersected by interfaces are enriched.

### 2.3. Implementation of Weibull Strength Distribution Model

The damage process in particle reinforced MMCs starts with the initiation of cracks at the locations of the particles either by the debonding of the matrix/particle interface or by the cleavage fracture of the particles. This is followed by crack growth in the matrix that leads to the ductile failure of the matrix ligaments between particles [37,51]. In this work, we focus only on incipient failure, and thus only the crack initiation by cleavage fracture of particles is considered. Due to the brittleness of the embedded particles, once a crack initiates, it is assumed to immediately propagate through the whole particle, leading to total splitting. The Weibull distribution model [29] has been widely used to predict the fracture probability of brittle particles in a ductile matrix [52,53]. The Weibull distribution function gives a simple but appropriate mathematical expression that can automatically account for the particle's size effect on their failure. The Weibull model is based on the weakest link statistics in

which the interaction between flaws can be neglected. A particle can be analogous to a chain consisting of several links and each link is analogous to a flaw in the particle. The chain fails as soon as a single link fails such that the failure probability of the chain is primarily dominated by the weakest link. If we want to calculate the probability of failure $(P_n)$ of a chain consisting of n links, and assume that the links are identical with the probability of failure $P$, then the probability of the chain not failing $(1 - P_n)$ will be equal to the probability that none of the links fail, i.e., $1 - P_n = (1 - P)^n$. The central idea of the Weibull distribution is to define a distribution function in an exponential form given by [29]:

$$P(X \leq x) = 1 - e^{-f(x)} \tag{12}$$

where, $f(x)$ can be any positive, non-decreasing function which should vanish at a value. The exponential form of the distribution function has intrinsic merits that account for the size effect since the cumulative failure probability can be easily denoted as:

$$(X \leq x) = 1 - e^{-nf(x)} \tag{13}$$

The cumulative Weibull probability can automatically account for the particle size effect. The critical flaw size decreases and the particle strength increase as the particle volume shrinks [54]. Therefore, the probability of fracture is high when the particle size and the stress acting on the particle increase [55]. The cumulative Weibull fracture probabilities were evaluated for each particle 'i' at each load increment by using the expression [36,53]:

$$P^i = 1 - \exp\left(-\frac{1}{V_0^i} \int_{\Omega_{\mathbb{P}_i}} \left(\frac{\sigma_1(x)}{\sigma_f}\right)^m d\Omega_{\mathbb{P}_i}\right) \tag{14}$$

where, $V_0^i$ is the reference volume of particle $\mathbb{P}_i$, $\sigma_1$ is the maximum principal stress at x, $\sigma_f$ and $m$ are the characteristic strength and Weibull modulus, respectively. The Weibull modulus is a measure of the degree of strength dispersion, i.e., large Weibull modulus narrows down the probability distribution [29]. To create a fracture plane in the particle $\mathbb{P}_i$, an on-plane point $(x_f)$ and the plane normal direction $(n_f)$ are needed and are given by:

$$x_f = \frac{\int_{\Omega_{\mathbb{P}_i}} x \cdot \sigma_1(x) \, d\Omega_{\mathbb{P}_i}}{\int_{\Omega_{\mathbb{P}_i}} \sigma_1(x) \, d\Omega_{\mathbb{P}_i}} n_f = \frac{\int_{\Omega_{\mathbb{P}_i}} n_1 \cdot \sigma_1(x) \, d\Omega_{\mathbb{P}_i}}{\|\int_{\Omega_{\mathbb{P}_i}} n_1 \cdot \sigma_1(x) \, d\Omega_{\mathbb{P}_i}\|} \tag{15}$$

XFEM method is accomplished in a C++ program. The overall algorithm can be summarized as Algorithm 1:

---

**Algorithm 1.** Particle fracture approximation

---

Identify all $\mathbb{P}_i \subset \mathbb{R}^3$ as intact particles

Assign a random critical probability $r_i \in [0, 1]$ to each particle $\mathbb{P}_i$

**for** Quasi-static load step i = 1 to n **do**

Calculate stress field with XFEM algorithm

**if** No intact particle left in the domain **then**

Continue to next step i = i + 1

**for** $\mathbb{P}_i \subset \mathbb{R}^3$ do

Compute Weibull fracture probability $P_f^i$

If $\max \left( P_f^i - r_i \right) > 0$ **then**

Break the particle $\mathbb{P}_i$ with the max $\left( P_f^i - r_i \right)$

Create a strong discontinuous plane at $x_f$ with normal direction $n_f$

Mark $\mathbb{P}_i$ as failed particle, $\mathbb{P}_i \not\subset \mathbb{R}^3$

Roll back and repeat load step i

else

Continue to next step i + 1

---

## 3. Fracture of SiC Particle Reinforced 2080 Aluminum Alloy by XFEM

To demonstrate the applicability of our XFEM algorithm for the fracture analysis of particle reinforced MMCs, in-situ uniaxial tensile testing was performed on the 20 vol.% SiC particle reinforced 2080 aluminum alloy (3.6% Cu, 1.9% Mg, 0.25% Zr) composite using X-ray synchrotron tomography, as discussed in Williams et al. [7,56]. The composite was prepared by the powder metallurgy process (Alcoa Inc., Alcoa, PA, USA), details of which has been provided elsewhere [57].

Electro discharge machining (EDM) was used to obtain dog-bone specimens of MMC with a gage length of 2.5 mm and a 0.75 mm square cross-section. Specimens were machined parallel to the extrusion axis. In-situ uniaxial tensile tests were carried out on these specimens in the synchrotron using the loading stage described in [56,58,59]. X-ray tomography was performed at the 2-BM beamline of the Advanced Photon Source (APS) at Argonne National Laboratory. The details of the tomography system at 2-BM have been described elsewhere [59,60]. The X-ray beam energy was approximately 24 keV. A LuAG:Ce scintillator screen was coupled with an objective lens and a CoolSnap K4 CCD camera to achieve a specimen pixel size of about 1.47 μm. 2D projections were collected at angular increments of 0.125° over a range of 180°. These 2D projections were then reconstructed using filtered back-projection algorithm.

### 3.1. Numerical Modeling

The domain dimensions of the specimen gage section were 750 μm × 750 μm × 2500 μm. As shown in Figure 1, the fractured plane in the experiment was close to one end of the gage section in the loading direction [56]. The images of the sample before and after fracture can be found elsewhere [56]. Note that the X-ray tomography was performed on a selected volume in the gage section, the position of which is shown in Figure 1. The selected volume is used as our simulation volume having the domain dimensions of 190 μm × 100 μm × 370 μm.

The SiC particles were modelled as exhibiting isotropic linear elastic response prior to fracture, with Young's modulus and Poisson's ratio of 410 GPa and 0.19, respectively [56]. A Weibull distribution (Equation (14)) can be used to estimate the fracture probability of SiC particles. The reference volume '$V_0$', characteristic strength '$\sigma_f$' and Weibull modulus '$m$' should obey $V_0 \sigma_f^m = \overline{V_0} \overline{\sigma}_f^m$ [52]. Gonzalez et al. [61] estimated these three parameters ($V_0$, $\sigma_f$ and $m$) for Al-SiC composite. The Weibull modulus '$m$', characteristic strength $\sigma_f$ and reference volume '$V_0$' (set as average particle volume) were estimated to be 6, 1323 MPa and $7.5^3$ μm$^3$, respectively. In this work, the average particle volume is 16,800 μm$^3$, the Weibull modulus is 6, and the characteristic strength is calculated to be 715 MPa

based on the relationship of the three parameters. The Young's modulus and Poisson's ratio of the aluminum matrix were taken as 74 GPa and 0.33, respectively [56]. The tensile stress-strain curve of the 2080-T6 aluminum alloy was taken from [62] and fitted according to a nonlinear $J_2$ plastic law as $\sigma_y(\gamma) = C\left(1 - e^{-b\gamma}\right) + \sigma_{yo}$, where $\sigma_y$ is the yield stress, $\gamma$ is the plastic strain, C = 185.1 MPa, b = 23.9 and the initial yield stress $\sigma_{yo}$ is 370 MPa. Figure 2 compares the fitted stress-strain curve with the experimental data, which shows an excellent agreement between the two.

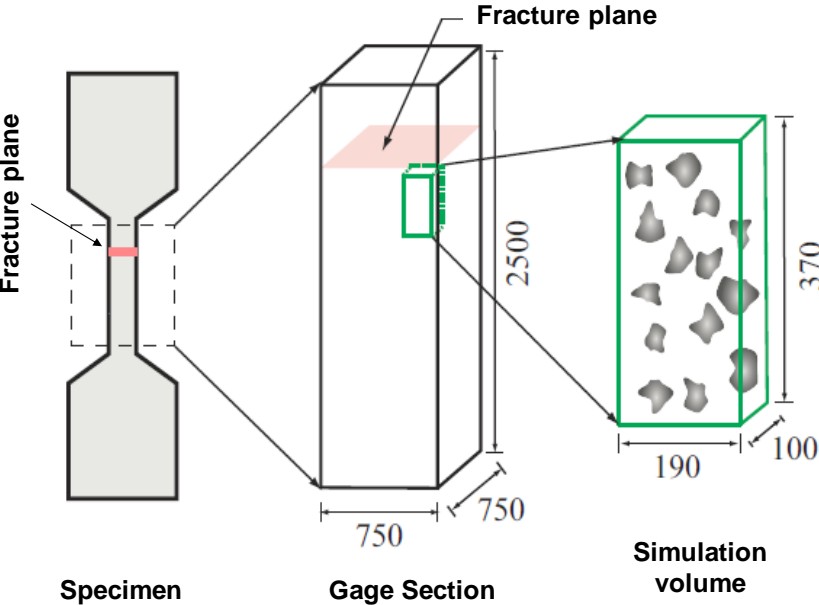

**Figure 1.** Location of the simulation domain relative to the gage section and fracture plane in the experimental specimen. All dimensions are in μm.

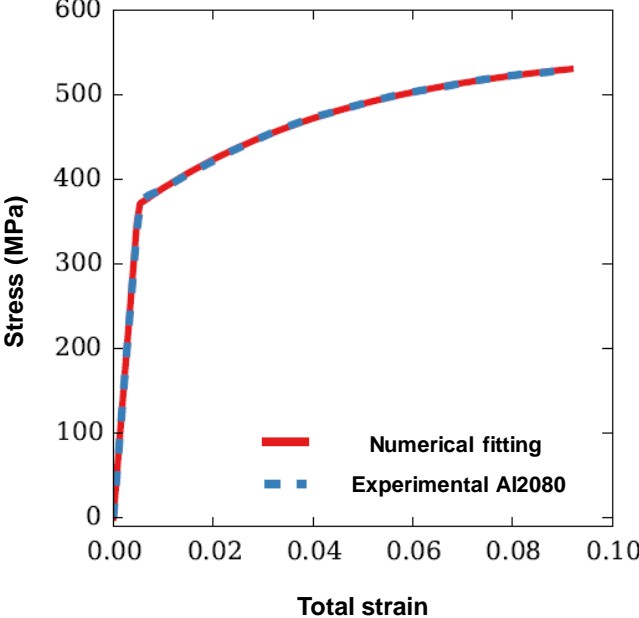

**Figure 2.** Comparison between the simulated and experimental stress-strain curves of 2080-T6 aluminum alloy.

### 3.2. Convergence study of Weibull Probability

The Weibull fracture probability of each particle is calculated by an integration involving the finite element approximation of the stress field as discretized by the elements inside and partially inside the particle. Since this is a norm-like measure of stress, one might expect that it should converge similarly to the energy error norm of the finite element approximation. To verify this hypothesis, we express the predicted Weibull fracture probability ($P$) as a summation of an exact probability $P^{exact}$ and a prediction error error $e(h)$:

$$P(h) = P^{exact} + e(h) \tag{16}$$

While the predicted Weibull probability and the error vary with the element size 'h', the exact Weibull probability is constant for a particle. Taking the derivative of Equation (16) with respect to element size, we have:

$$\frac{dP(h)}{dh} = \frac{de(h)}{dh} \tag{17}$$

In the limit as the element size decreases to zero, we assume that the error follows a power law, and thus in a log-log scale, the relationship between the rate at which the predicted failure probability decreases with element size can be written as:

$$\log\left(\frac{dP(h)}{dh}\right) = \beta + \alpha \log_{10}(h) = \log_{10}\left(10^\beta h^\alpha\right) \tag{18}$$

Which is a linear equation with $\alpha$ and $\beta$ being the slope and the y-axis intercept, respectively. By combining Equations (17) and (18), the probability error can be estimated as:

$$\frac{de(h)}{dh} = 10^\beta h^\alpha \tag{19}$$

$$e(h) = \frac{10^\beta}{\alpha + 1} h^{\alpha+1} \tag{20}$$

Therefore, the convergence rate of the fracture probability is $\alpha + 1$, where $\alpha$ is the convergence rate of the derivative of fracture probability. Four particles with different geometries were used to study the convergence of $dP/dh$, the X-Y views of which are shown in Figure 3. Figure 3a shows a spherical particle (ideal geometry) located at the center of a cube having length of 60 μm. The ratio of the particle's radius over the cube length is 0.3 and the particle volume is 24,429 μm$^3$. Figure 3b–d are three particles with realistic geometries. These three particles are shifted to the center and the domain lengths are chosen such that the same padding distance of matrix is maintained in the three orthogonal directions. The volume of the three particles are 27,143, 23,570 and 16,424 μm$^3$, respectively. For all cases, only rigid body motions are fixed and traction is applied in Y direction. The applied traction is increased linearly from 0 MPa to 400 MPa in four load steps. Here, oct-tree mesh refinement is used and elements close to a material interface or inside a particle are refined with a single iteration of refinement.

Figure 4b shows the fitted convergence rates for the four particles. The convergence rates of $dP/dh$ are close to 1, indicating that the error in the computed fracture probability decreases quadratically with respect to the element size, which is one order higher than the optimal, linear rate of convergence of the energy error norm for 8-node hexahedral elements. Figure 4c shows the relative probability error w.r.t. the element size used for particles. The relative error is calculated by the ratio of $e(P)$ over the fracture probability when the element size is 0.5 μm. It is evident that the particles with the realistic geometries (Figure 4b–d) have higher probability error than the particle with spherical geometry (Figure 4a) with the same element size. When the element size is close to 1 μm, the probability error is less than 10%.

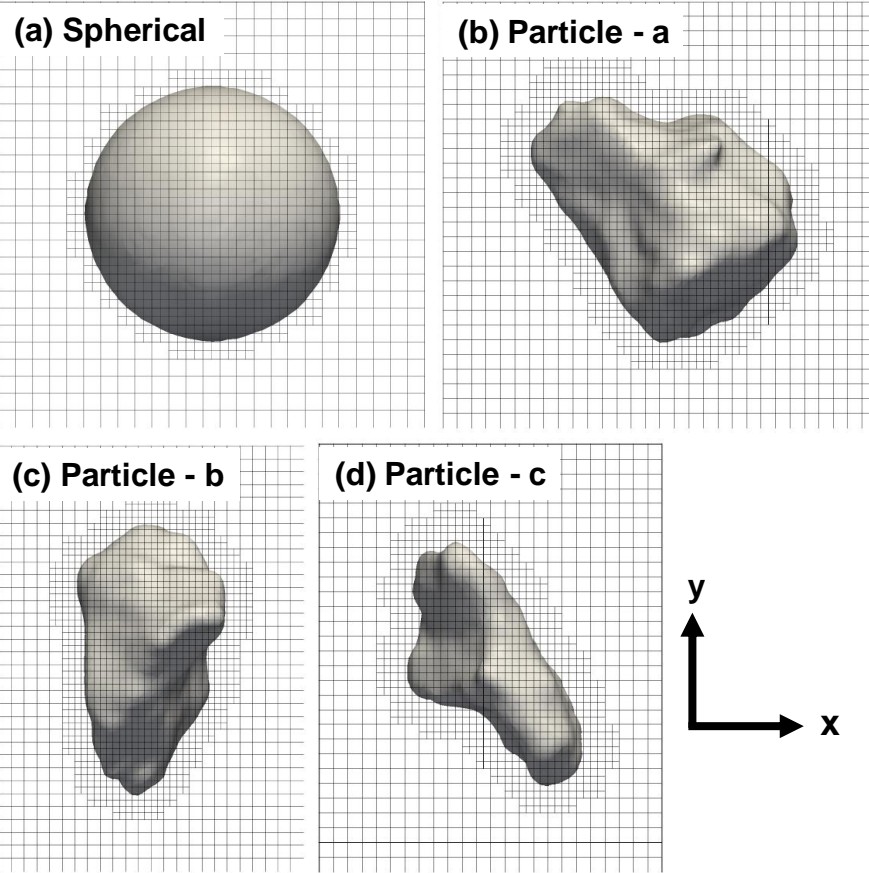

**Figure 3.** X-Y view of particles in mesh wireframe used in the convergence study.

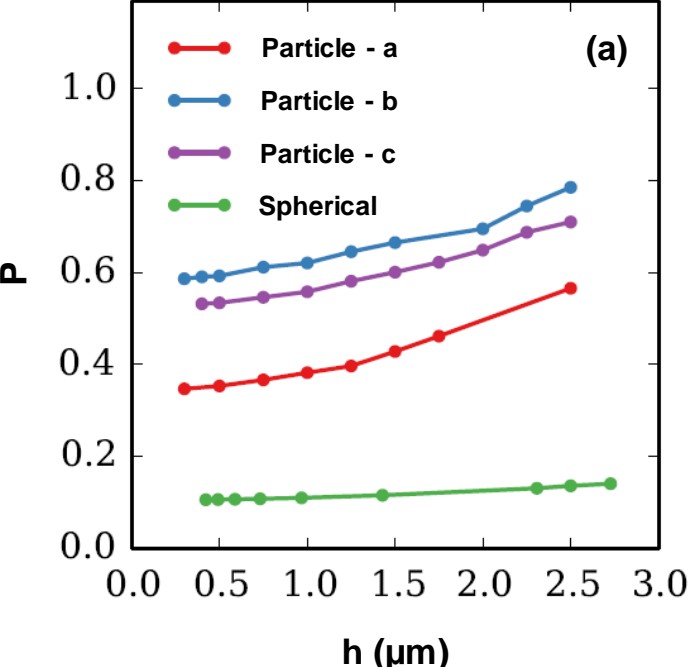

**Figure 4.** *Cont.*

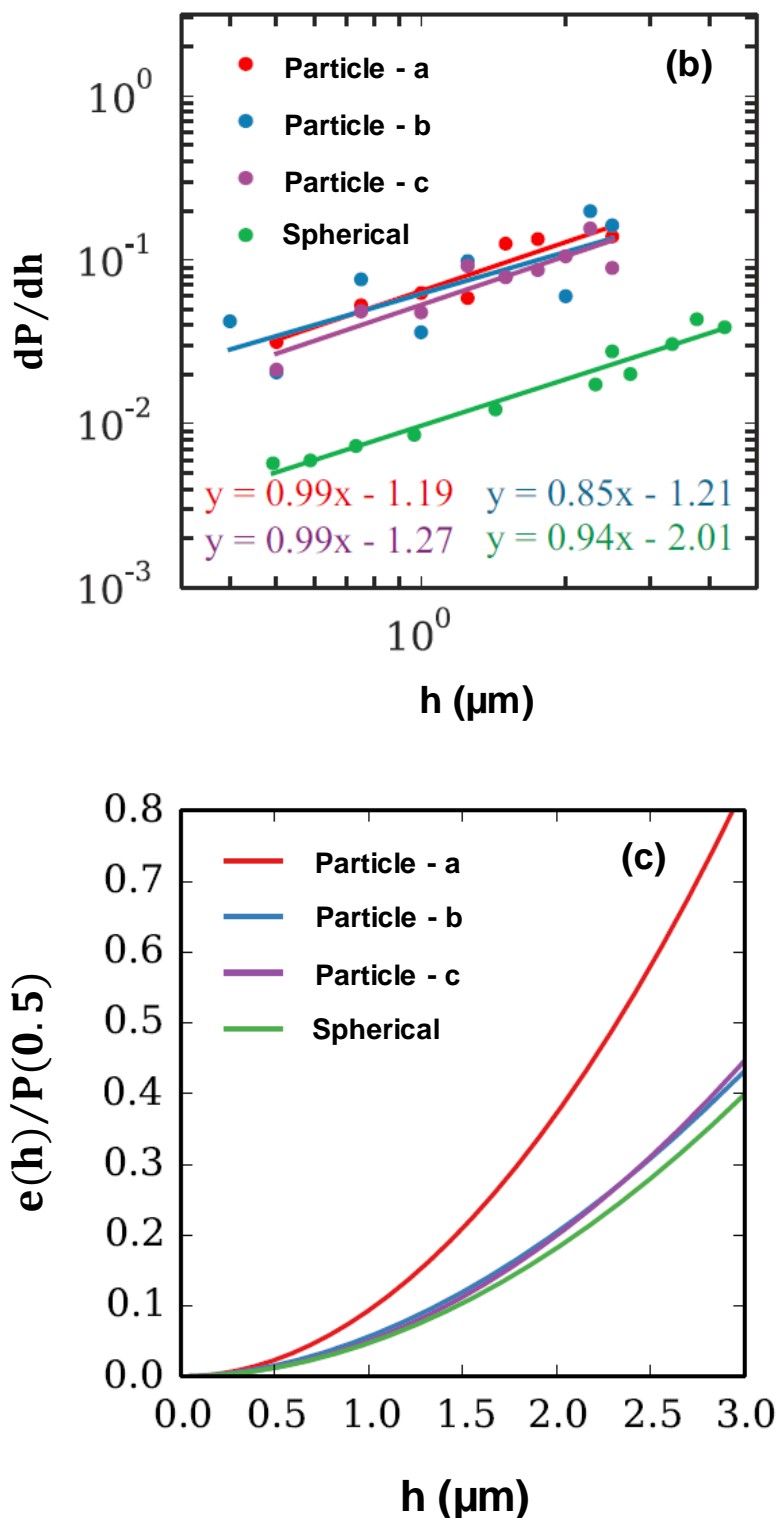

**Figure 4.** (**a**,**b**) Convergence of the derivative of Weibull fracture probability in particles with different geometries, and (**c**) corresponding estimated probability error.

### 3.3. Fracture Analysis of the Simulation Volume

Figure 1 shows the domain size of the simulation volume and its relative position in the experimental specimen. Since the simulation volume is from the boundary of the gage section, symmetric boundary conditions have been applied in the fracture analysis and the traction is applied

on the top surface of the simulation volume, as shown in Figure 1. Quasi-static analysis is performed and the traction is increased linearly from 0 to 405 MPa in 18 steps. Oct-tree mesh refinement applied to an initially uniform mesh of 19 × 10 × 37 elements, where elements that are intersected by a material interface or within particles are split into 8 child elements. After three such refinement iterations, the simulation volume is discretized using 560,226 elements, with a minimum element size discretizing the particles and their interface is 1.25 μm. Of the initial 41 SiC particles in the simulation volume, 31 fractured by the final load step. Figure 5a shows the 41 particles embedded within the mesh of the simulation domain, where the cleavage planes of the fractured particles are shown as black curves on the particle surfaces. Figure 5b shows the axial stress distribution on the particles, where blue color stands for zero stress (corresponding to crack surfaces) and the maximum axial stress is scaled to 1 GPa. The normal direction of fracture surfaces is computed by Equation (15) and the numerical normal directions are aligned nearly with the loading direction. Figure 5c,d show contours of the effective strain and axial stress fields on a cut plane of the domain at the end of the final load step, respectively.

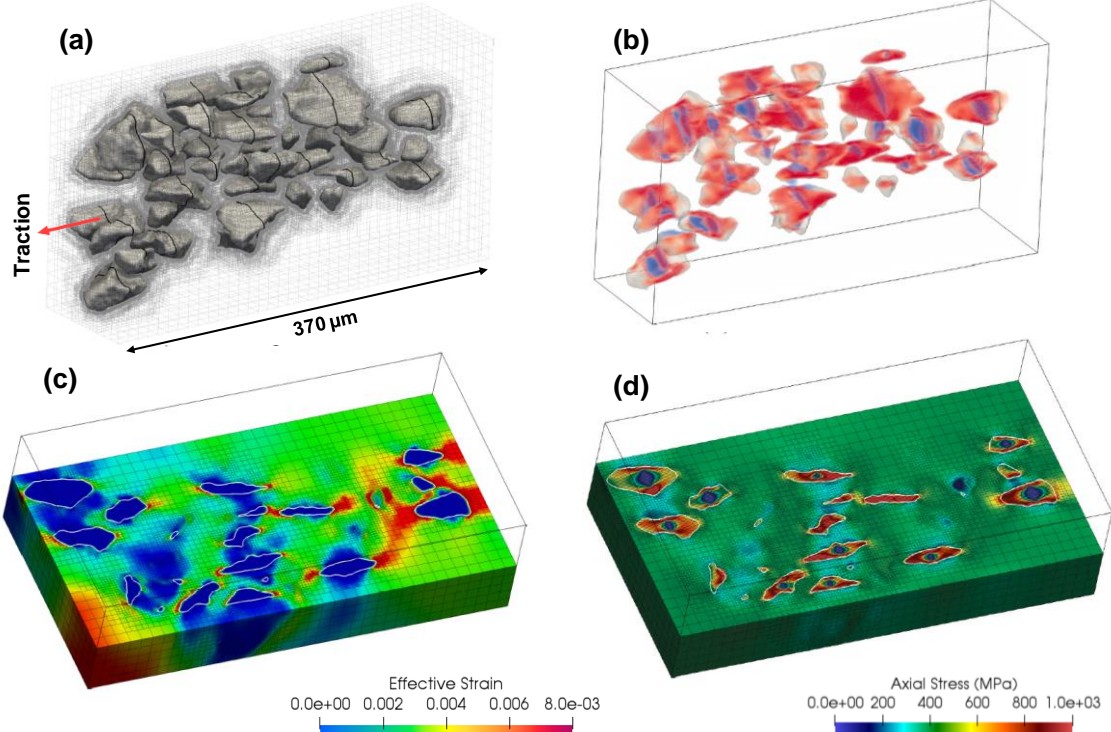

**Figure 5.** (**a**) Particle geometries in the oct-tree mesh wireframe. Fractured surfaces are denoted as black lines, (**b**) axial stress distribution on the particles, (**c**) effective strain on a slice parallel to the loading direction. Note that the traction is applied on left surface and the maximum effective strain is scaled to 0.008, and (**d**) axial stress on a slice parallel to the loading direction. The maximum axial stress is scaled to 1 GPa.

Figure 6 shows the comparative plot of experimental and simulated stress-strain curves of the composite. In the experiment, the onset of damage in the specimen was observed to begin close to 440 MPa [56]. In the numerical model, the same simulation volume and mesh scheme were used; however, the traction was increased from 0 to 450 MPa in 20 steps. It is clear from Figure 6 that there is a good agreement between the simulated and experimental stress-strain curves, however the simulated stress is slightly larger than the experimentally measurements yielding, which can be attributed to several assumptions made in the model. First of all, only crack initiation by cleavage fracture of particles is considered, whereas the particle and matrix are considered to be perfectly bonded and the matrix is idealized without fracture. However, in the experiment, localized void growth are observed

which can increase the strain under the same traction [7]. These factors are likely the reason for the slightly higher traction values in the simulation after yielding.

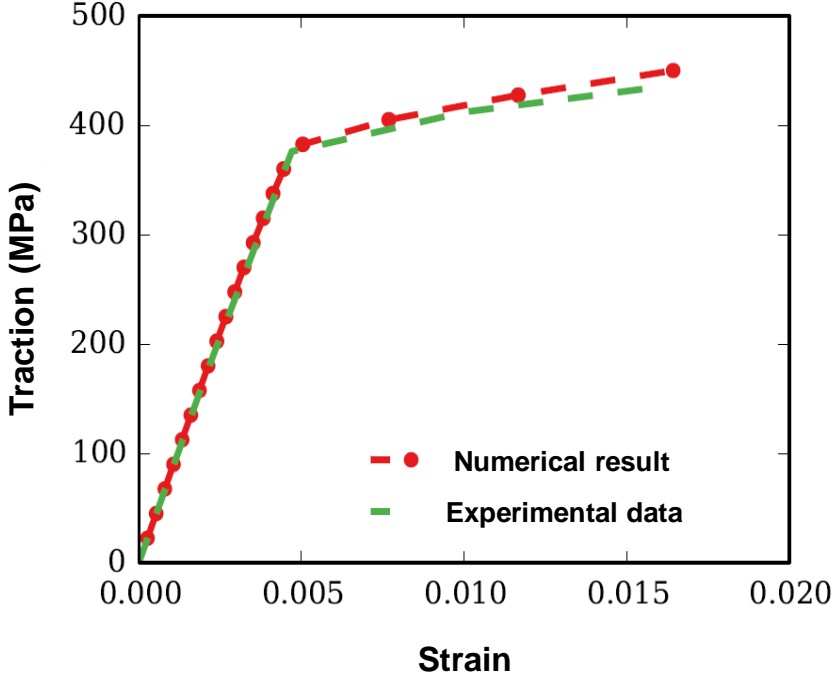

**Figure 6.** Comparison of numerical and experimental strain-stress curves for the composite.

There are 41 particles in the simulation volume and the particle volume ranges from 1421 to 36,350 $\mu m^3$. As the Weibull fracture model can account for the particle size effect automatically, large particles tend to fracture before small particles. Figure 7 shows the average volume of fractured particles with the applied traction. It is evident from Figure 7 that the particles start to fracture as soon as the applied traction reaches to 247.5 MPa. As the applied traction increases, smaller particles fracture, leading to a decrease in the average volume of the fractured particles (Figure 8).

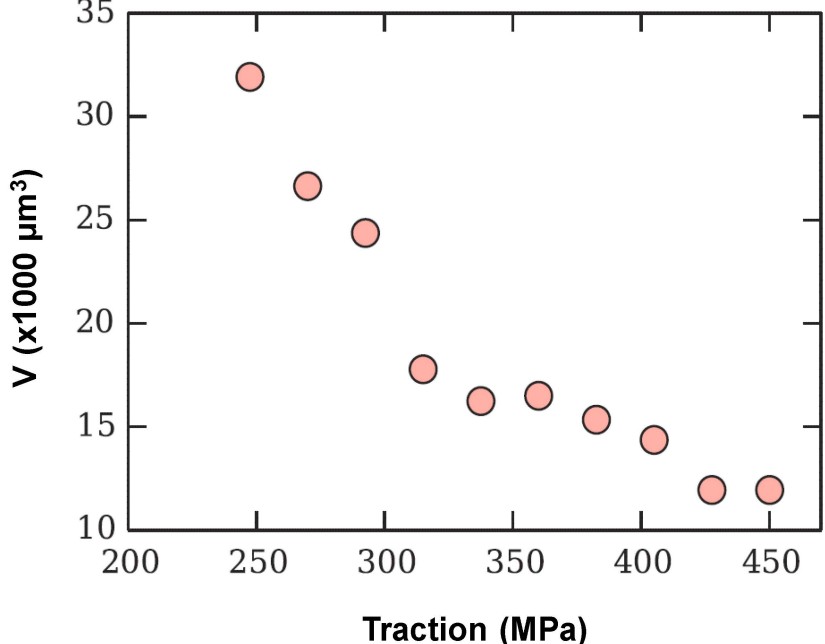

**Figure 7.** The average volume of the fractured particles in each load step during simulation.

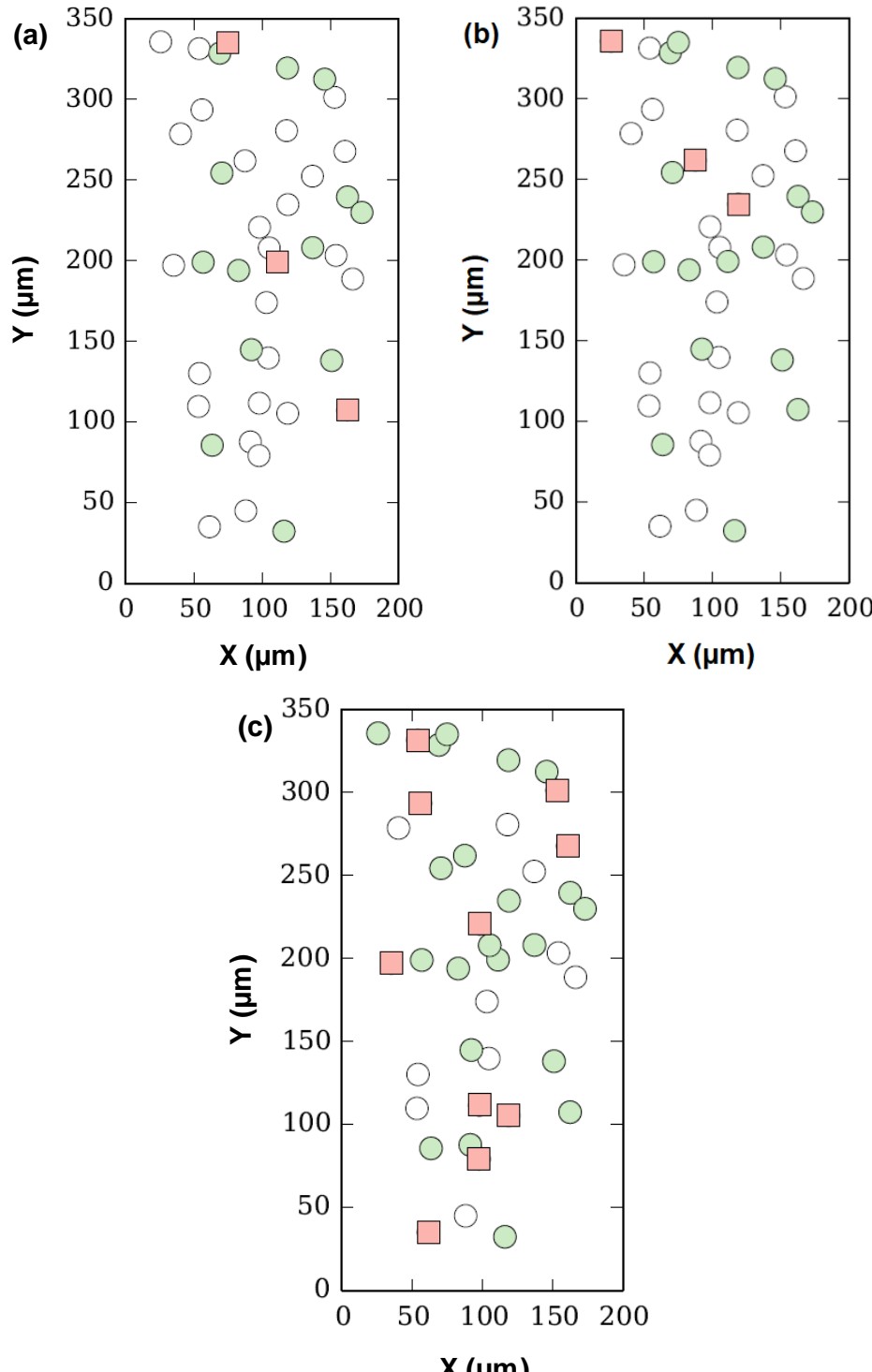

**Figure 8.** Graphical representation of the relative position of centroids of SiC particles on a projected XY plane when (**a**) Traction is 337.5 MPa, (**b**) Traction is 360 MPa, and (**c**) Traction is 405 MPa. Solid circles in white represent particle centroids that are not cracked. Green circles and red squares are particle centroids that are "previously" cracked and "newly" cracked, respectively, relative to the previous load step.

Although Figure 5a shows all the fractured particles in 3D, it is not straightforward to visualize their relative positions as one particle can interrupt the view of others. Therefore, in order to distinctly visualize the relative positions of the fractured particles, the centroids of all the particles were projected

to a x-y plane, in which the y-direction is along the loading direction. Figure 8 shows the centroid positions of the fractured particles on a projected X-Y plane under three load steps, i.e., 337.5 MPa (Figure 8a), 360 MPa (Figure 8b) and 405 MPa (Figure 8c). Note that each marker in Figure 8 corresponds to the centroid of a single particle and all particles have been split into three categories: (i) intact, i.e., particles which are not fractured (white circle), (ii) particles fractured in previous load steps (green circles), and (iii) particles fractured in the current load step (red square). Although the sequence of cracking of particles does not appear to follow a straightforward order due to the randomness of critical fracture probability assigned to each particle, some patterns can still be observed. Particles tend to fracture in groups and particles adjacent to previously cracked particles are more prone to fracture. This can be attributed to the redistribution of load in the composite. Once a particle fractures, the stress field is redistributed in its surrounding region, which results in increased load carried by the near-by particles, making them more prone to fracture.

In order to study the influence of one cracked particle on others, the change in the fracture probability ($\Delta P$) of the intact particles were calculated before and after the fracture of a single particle. Figure 9 shows the centroids of particles, along with their $\Delta P$, projected on a XY plane. Figure 9a–d show the distribution of $\Delta P$ of the centroids of un-cracked particles after four different particles have fractured. Note that the newly cracked particles are denoted as yellow circle with a black line, white circles are previously cracked particles and rest are un-cracked/intact particles. It is clear from Figure 9 that the fractured particle can influence the fracture probability of the near-by particles. The fracture probability tends to increase for a particle that are close to the newly cracked particle and are on the same XZ plane (perpendicular to the loading direction). This tendency is straightforward to understand because as soon as one particle breaks, the remaining particles on the same XZ plane have to support extra stress.

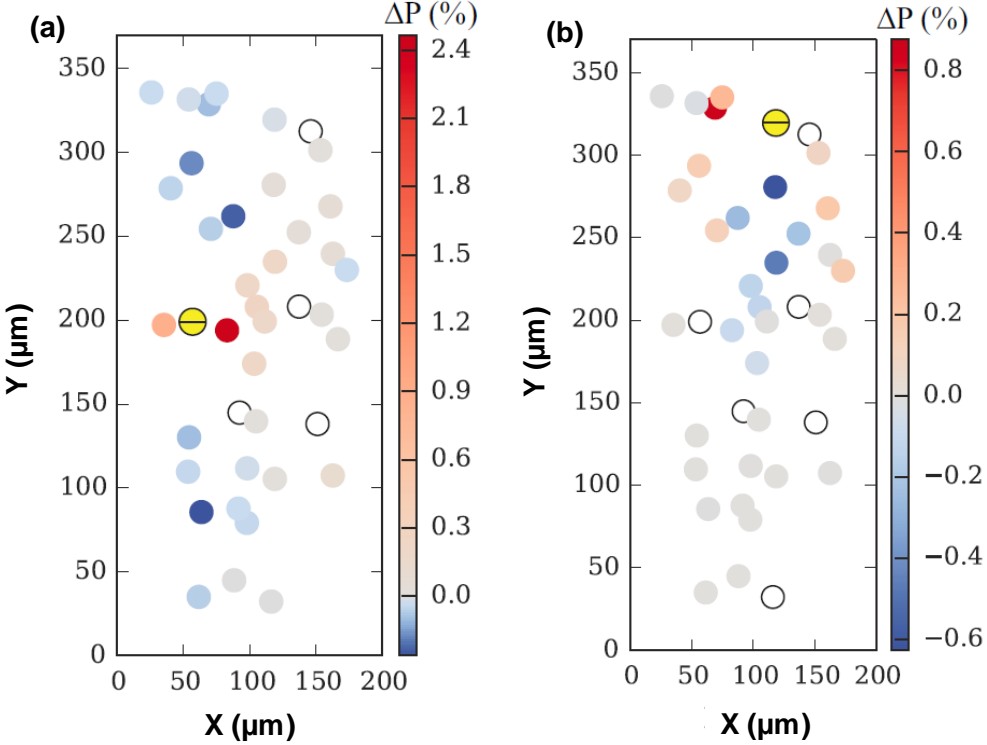

**Figure 9.** *Cont.*

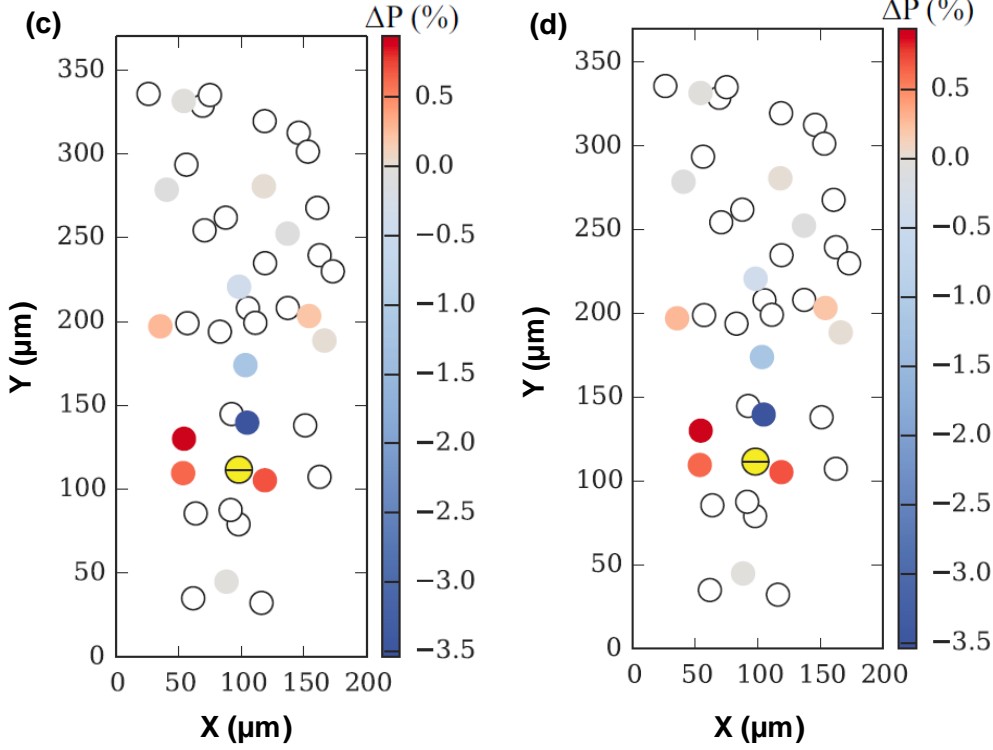

**Figure 9.** (**a**–**d**) Shows the distribution of ΔP of the centroids of un-cracked particles after four different particles have fractured. Previously cracked particles are denoted in solid white circles. The newly cracked particle are denoted in a solid yellow circle with a black break surface. Un-cracked particles are colored by the change of Weibull fracture probability before and after fracture.

While the fracture probability is inclined to decrease on a particle that is not far away to the newly cracked particle but their relative position is parallel to the loading direction. This tendency is caused by the change of stress flow before and after fracture. Before fracture, the top and bottom parts of the particle in the loading direction tend to have higher stress than the middle part. After the fracture, the stress flows to the crack tip and particles usually break around a cross section in the middle of the loading direction. Since the stress unloading of the top and bottom parts, the adjacent particles close to the top and bottom parts tend to withstand more strength.

In order to evaluate the influence of particle geometry on the particle fracture probability, Weibull fracture probability versus applied traction was studied for different particle geometries (Particle-m, Particle-n and Particle-p, as shown in Figure 10). In this case also, each particle is shifted to the center of the simulation volume and the domain sizes are set to maintain a constant fill ratio for all cases. As shown in Figure 11a, the Weibull probability increases with traction much more rapidly for the real particles. Further, the three realistic particles exhibit much higher Weibull probability compared with sphere (ideal particle) when traction is higher than 200 MPa, indicating that the aspect ratio of the particles plays an important role on their fracture behavior. Note that the Particle-m and spherical particle exhibit the highest and lowest aspect ratio, respectively. The particle aspect ratio is calculated by the ratio of the longest Feret distance over the shortest Feret distance. Particle-m has a narrow volume and its fracture probability is the highest among all the studied particle geometries at the same traction. The lowest Weibull probability was observed for the spherical particle indicating that the particles with the smallest aspect ratio (i.e., 1 for sphere) are less prone to fracture. In order to further understand the influence of particle geometry on the fracture probability, the Weibull probabilities about one hundred particles, having a range of aspect ratio, were calculated at the applied traction of 400 MPa. The particle volumes range from 5192 $\mu m^3$ to 28,350 $\mu m^3$. As shown in Figure 11b, particles are separated into three groups with equal volume interval. It is clear that larger particles exhibit

higher fracture probability and the slopes of the fitted curves for the three volume intervals are close to each other. This is attributed to the higher probability of the presence of the Griffith's crack in the larger particles. In addition, SiC particles with larger aspect ratios show higher Weibull probabilities than the particles with lower Weibull probabilities indicating that the particles exhibiting higher aspect ratios are more prone to fracture. Overall, the larger particles with higher aspect ratios exhibit higher Weibull probabilities and are more prone to fracture, which is in complete agreement with the observations made in experiment [7].

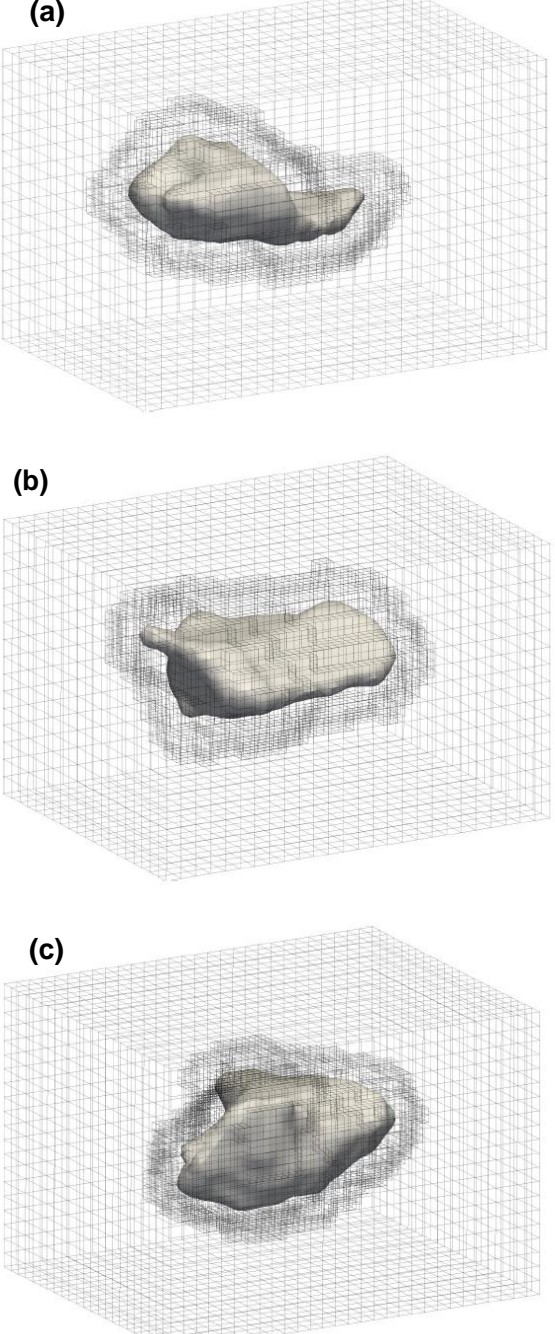

**Figure 10.** Particle geometries in mesh wireframe that are used in history plot of fracture probability (**a**) Particle-m, (**b**) Particle-n, and (**c**) Particle-p. Rigid body motions are fixed.and traction is applied on both left and right surfaces.

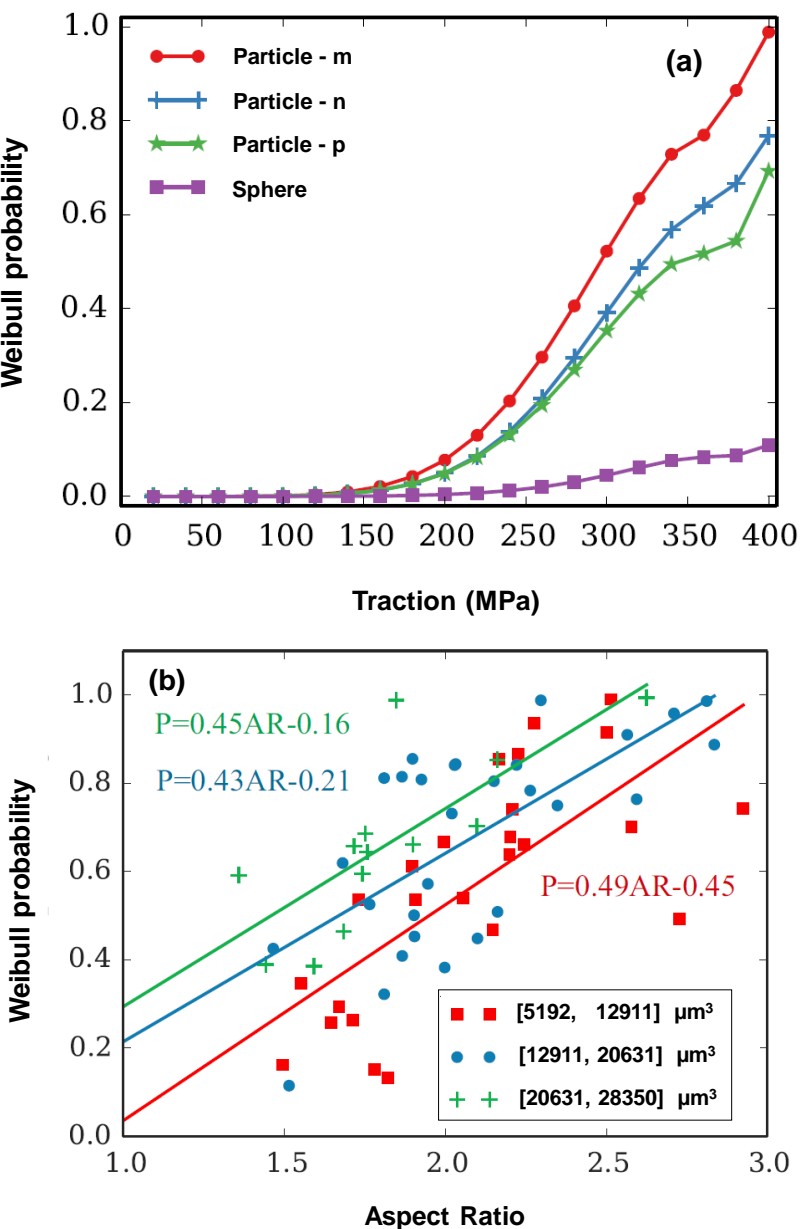

**Figure 11.** Characteristic study of the influence of particle geometry on the magnitude of Weibull fracture probability (**a**) history plot of fracture probability, (**b**) fracture probability vs. aspect ratio.

## 4. Conclusions

In this study, the Weibull strength distribution model has been implemented in the XFEM algorithm for the modeling of particle fracture in metal matrix composites (MMCs) with realistic particle geometries initialized from X-ray tomographic data of 20 vol.% SiC particle reinforced 2080 aluminum alloy composite. The simulated tensile stress-strain curve was found to be in good agreement with the experimental stress-strain curve obtained from *in situ* tensile testing. The Weibull modulus '*m*' and characteristic strength '$\sigma_f$' were estimated to be 6 and 715 MPa, respectively. The average volume of the fractured particles was found to decrease with the applied traction. Fracturing of a particle resulted in an increase in the fracture probability of the neighboring particles on the same plane. In addition, the Weibull fracture probability was observed to be influenced by both the particle volume and shape, i.e., larger particles with higher aspect ratios were more prone to fracture.

**Author Contributions:** Conceptualization, N.C., J.O.; methodology, R.Y., S.S.S., X.L.; formal analysis, X.X.; investigation, R.Y., S.S.S., X.L.; data curation, R.Y., S.S.S.; writing—original draft preparation, R.Y., S.S.S.; writing—review and editing, N.C., J.O.; supervision, N.C., J.O.; funding acquisition, N.C. All authors have read and agreed to the published version of the manuscript.

**Funding:** This research was funded by Air Force Office of Scientific Research through the multi-university research initiative (MURI) (Dr. Ali Sayir, program manager).

**Acknowledgments:** NC, SSS, and RY are grateful for financial support from the Air Force Office of Scientific Research through the multi-university research initiative (MURI) (Ali Sayir, program manager). The authors also acknowledge the use of facilities at the Center for 4D Materials Science (4DMS) at Arizona State University.

**Conflicts of Interest:** The authors declare no conflict of interest.

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
