# Peer review of "Fracture Analysis of Particulate Metal Matrix Composite Using X-ray Tomography and Extended Finite Element Method (XFEM)"

_jcs, doi:10.3390/jcs4020062_

Round 1

Reviewer 1 Report

line 168: in the equation on the right side in the brackets shouldn't it be an uppercase 'P'?

lines 266, 267: reference should be to Fig. 3.

line 292: which symmetric boundary conditions have you applied since your simulation volume is not in the middle of the specimen (Fig. 1)?

line 301: 'shown as black curves'

line 314, 315: you mention an onset of damage citing [55]. But the stress/strain curve in [55] is a different one than the one you are showing in this paper. In the sentence before you mention Fig. 6 with the presented stress/strain curves. How does this fit together?

line 363: 'that are close to'

line 389: 2x 'feret distance'

Reviewer 2 Report

The authors investigated the use of extended finite elements method (XFEM) and X-ray synchrotron tomography to study fracture mechanisms in aluminium matrix composite. The paper is well executed. However, the following comments need to be addressed.

There is research that carried out in 2006 that uses X-ray tomrogorhy and FEA to simulate mechanical properties of metal composites. It is important that you added to the literature and emphasis on what novel aspect you are introducing in this paper

  1. G. Watson Peter D. Lee R. J. Dashwood R Philippe Young “Simulation of the mechanical properties of an aluminium matrix composite using X-ray microtomography” Metallurgical and Materials Transactions A 37(3):551-558

20 vol.% SiC particle reinforced 2080 aluminium alloy composite, why 20%? Are there particular applications, benefits?

Include the software used in the FEA

L 15:  fracture mechanisms in these materials: under tensile loading?

L17: composite. composite remove the repeated word

L157: leading to total splitting. While the particle and matrix are assumed to be always 157 perfectly bonded, the cracks are assumed to be arrested at the particle-matrix interface. Not clear what assumption did you use. splitting particulate or propagation through the interface?

L212: In-situ uniaxial tensile tests were carried out on these specimens in the synchrotron. Include the setup photo of the samples inside the tensile testing and the synchrotron.

What equipment, manufacturer did you use in tensile testing? Is it the same machine as the synchrotron.

Include a photo of the sample before and after a fracture.

Figure 2 is it your generated figure? Did you use the modelling and experimental testing of 2080-T6 without particulates?

What are a,b,c,d in figure 9? Define them in the caption and text.

Round 2

Reviewer 1 Report

The paper is fine for me with the corrections included.

Reviewer 2 Report

The authors addressed all the comments and the paper can be published in the present form